# Do Spanish Triathletes Consume Sports Supplements According to Scientific Evidence? An Analysis of the Consumption Pattern According to Sex and Level of Competition

**DOI:** 10.3390/nu15061330

**Published:** 2023-03-08

**Authors:** Rubén Jiménez-Alfageme, José Miguel Martínez-Sanz, David Romero-García, Daniel Giménez-Monzo, Samuel Hernández Aparicio, Antonio Jesús Sanchez-Oliver, Isabel Sospedra

**Affiliations:** 1Faculty of Health Sciences, University of Alicante, 03690 Alicante, Spain; 2Food and Nutrition Research Group (ALINUT), University of Alicante, 03690 Alicante, Spain; 3Physiotherapy Department, Faculty of Health Sciences, European University of Gasteiz—EUNEIZ, 01013 Vitoria-Gasteiz, Spain; 4Nursing Department, Faculty of Health Sciences, University of Alicante, 03690 Alicante, Spain; 5Department of Community Nursing, Preventive Medicine and Public Health and History of Science Health, University of Alicante, 03690 Alicante, Spain; 6Departamento de Motricidad Humana y Rendimiento Deportivo, Facultad de Ciencias de la Educación, Universidad de Sevilla, 41004 Sevilla, Spain

**Keywords:** sports supplements, triathlon, sport nutrition, scientific evidence, sport performance

## Abstract

Background: The use of sports supplements (SS) to improve sports performance is very common in athletes. In the case of triathletes, the physiological characteristics of the sport may require the use of certain SS. Although the consumption of SS is widespread in this sport, very few studies have investigated it thus far. The aim is to analyze the pattern of SS consumption by triathletes according to sex and the competitive level. Methodology: This is a descriptive cross-sectional study on the consumption and habitual use of SS of 232 Spanish-federated triathletes. Data were collected through a validated questionnaire. Results: Overall, 92.2% of the athletes consumed SS, but no significant differences were found in terms of competition level or sex. Yet, significant differences were found regarding the level of competition for total SS (*p* = 0.021), the total number of Group A supplements from the AIS classification (*p* = 0.012), and for the ergogenic aids (*p* = 0.003). The most-consumed SS were bars, sports drinks, sports gels, and caffeine (83.6%, 74.1%, 61.2%, and 46.6%, respectively). Conclusions: The consumption of SS by triathletes is high, and the number of SS consumed rises from the regional to the national and international levels. The four SS most consumed were included in category A of the AIS (greatest scientific evidence).

## 1. Introduction

The triathlon is an endurance and individual sport included in the Olympic Games since the year 2000 [1] and has grown in popularity since its appearance in 1974. It is characterized by having three sports disciplines: swimming, cycling, and running, which are carried out consecutively, with the aim of completing the competition in the shortest possible time [2]. This sport offers a wide range of event formats, ranging from the mixed relay race, which lasts about 20 min, to the sprint distance race, which lasts approximately one hour, and the long-distance triathlon (Ironman), which takes place over a period of 8 to 9 h at the elite level. In addition to the high training volumes typically required for endurance sports, training for three different sporting disciplines simultaneously requires a large number of training sessions every week [3,4]. Training at such high volumes can increase the likelihood of illness and injury. Nevertheless, recent advancements in knowledge in this area can minimize these risks while simultaneously optimizing performance [5].

Sports performance in triathlons depends on factors such as anthropometric characteristics, physiological capacity, technical skills, and the competition strategy carried out by the triathlete [6]. This sport places high physical and physiological demands on the athlete [7,8]. In addition, factors such as carbohydrate (CH) intake, hydration, and adverse results related to nutrition and acclimatization of the athlete are determining factors [9,10]. Therefore, it is not uncommon for triathletes to resort to sports supplements (SS) to complement their diet and enhance their performance during competitions, especially in long-term competitions [11,12]. Triathletes typically compete in a self-sufficient or semi-self-sufficient regimen, which requires them to bring their own food and hydration, sometimes with the assistance of supplies provided by organizers. This is one reason why the use of SS can be advantageous, as it allows them to optimize space and weight in their sports equipment [13]. Training sessions are usually conducted in a self-sufficient or semi-self-sufficient manner depending on the duration and whether the triathlete has external support [12,13].

SS can be defined as any food, food component, nutrient, or non-food component intentionally ingested as part of a regular diet to achieve a specific effect on health or performance [9]. The Australian Institute of Sport (AIS) [14] categorizes SS into four groups (A, B, C, and D) based on the degree of scientific evidence. Group A comprises SS compatible with use in specific situations in sports, using evidence-based protocols. Group B includes SS requiring further investigation and that should be used cautiously. Group C includes SS without evidence of beneficial effects. Lastly, Group D includes prohibited substances or substances with a high risk of contamination that could result in a positive anti-doping test.

Currently, between 30 to 100% of elite and amateur athletes consume SS to improve their sports performance [10,11,15,16,17,18,19,20,21,22]. The most significant factors that influence SS consumption are the kind of physical activity performed, the athlete’s level of competitiveness, and sex. In particular, SS consumption tends to be higher among athletes engaged in more competitive levels of sport and among male athletes (in the same sport) [21,22]. Among the most-consumed SS, we find products such as carbohydrate drinks, protein bars, sports gels, caffeine, whey protein, creatine, or branched-chain amino acids (BCAAs) [10,11,15,16,17,18,19,20,21,22].

Although the bibliography is scarce, some studies can be found on the use of SS in endurance sports, specifically in mountain runners and open-water swimmers [20,21,22], with these endurance athletes consuming SS such as sport bars (81.9–62.9%), sports drinks (75–60.5%), caffeine (39.4–48.6%), and sports gels (52.9%). On the other hand, other works have analyzed the nutritional intake in these types of competitions [23,24,25], but to date, no research has been carried out exclusively in triathletes to study the prevalence and consumption patterns of SS. Therefore, the aim of this study was to analyze the pattern of SS consumption by triathletes by studying the differences based on sex and the competitive level (regional, national, and international). Regarding the main hypotheses of this work, the following are proposed:

**Hypothesis** **1** **(H1).**
*The type of supplements that are most consumed are similar according to the sex of the athletes.*


**Hypothesis** **2** **(H2).**
*The total consumption of SS by athletes at a higher competitive level is higher.*


**Hypothesis** **3** **(H3).**
*Differences in the consumption and type of SS are not expected between triathletes and other endurance athletes such as swimmers and mountain runners.*


## 2. Materials and Methods

### 2.1. Type of Study

This is a descriptive and cross-sectional study on the consumption and regular use of SS by Spanish triathletes. The sample size calculation was performed with Rstudio software (version 3.15.0, Rstudio Inc., Boston, MA, USA). The significance level was set a priori at *p* = 0.05. The standard deviation (SD) was set according to the total SS data from previous studies on elite Spanish athletes (SD = 2.1) [26]. With an estimated error (d) of 0.27, the sample size needed was 232 subjects. The study population was selected by non-probabilistic, non-injury, convenience sampling among triathlon federations and triathlon clubs throughout Spain.

### 2.2. Participants and Sample Size

The sample consisted of 232 triathletes who were aged 34.79 ± 9.93 years old. Among them, 165 were men and 67 were women, and all were of legal age. The participants had competed in regional, national, and international competitions for at least two years and had not suffered from any injuries or illnesses in the six months prior to the survey. The participants’ competitive level varied from regional competitions (held at the provincial and regional level) to national competitions (held throughout Spain) and international competitions (held worldwide). Table 1 provides information on the age, basic anthropometric characteristics, and years of sports experience of the study participants.

### 2.3. Instruments

The study utilized a questionnaire that had been previously used in similar studies [17,19,20,21,22,27]. The selected supplement consumption questionnaire was validated for content, applicability, structure, and presentation [28]. It contains a total of 35 questions divided into three main sections. The first section collects anthropometric data such as age, weight, height, as well as personal data such as sex and social data such as the autonomous community of residence. This section has six questions. The second section includes nine questions that cover the practice of sports and its context, such as years of practice and the number of competitions. The last and most extensive section focuses on diet and the consumption of supplements. It includes 20 questions, such as what supplements the respondents consumed, why they consumed them, who advised them, where they purchased them, when they took them, and their perception of the results after consumption. The questionnaire can be viewed in the Appendix A.

### 2.4. Procedure

To select the sample for this study, we contacted via email the representatives of each regional triathlon federation in Spain as well as triathlon clubs registered with these federations. We informed them about the study’s characteristics and requested their collaboration. Once they agreed to participate, we sent them an email containing a link to the supplement consumption questionnaire, which the triathletes could complete voluntarily, electronically, and anonymously. The protocol adhered to the Declaration of Helsinki for human research and was approved by the ethics committee of the University of Alicante with file number UA-2021-02-01.

### 2.5. Statistical Analysis

To verify whether the variables had a normal distribution, a Kolmogorov–Smirnov test was applied, and Levene’s test was used to verify homoscedasticity. The quantitative data obtained are presented as mean (M) + SD, while the qualitative variables are expressed as percentages and frequencies. An ANOVA was performed according to sex (male, female), level of competition (regional, national, and international), and the interaction between sex and level of competition to analyze the differences in the total consumption of SS as well as the SS consumed from the different categories defined by the AIS [14]. When significant differences were found between groups, a pairwise comparison was performed using the Bonferroni correction for multiple comparisons. Regarding the analysis of the athletes who consumed SS, the reason for consumption, the place where they obtained them, and who advised them to consume them, a chi-square (X2) test was used to verify the existence or not of differences between athletes of different sex and level of competition. As for the SS that were consumed by at least 10% of the sample, a chi-square test (X2) was also performed to verify possible differences according to sex or level of competition. The level of statistical significance was established as *p* < 0.05. The statistical analysis was carried out with the Statistical Package for Social Sciences v.20 software for Windows (SPSS) (IBM, Armonk, NY, USA).

## 3. Results

From the total sample, 92.2% declared consuming SS. Regarding the level of competition, no significant differences were found (F = 0.637; *p* = 0.727) regarding the consumption of supplements, showing that 91.6%, 92%, and 96.2% of regional, national, and international triathletes, respectively, consumed SS. Although the percentages of consumption in all three categories were similar, there was an increased trend between regional, national, and international athletes. Regarding sex, no significant differences were found either (F = 0.952; *p* = 0.329), with the results indicating that 93.3% of men and 89.6% of women consumed SS.

Regarding the response of the participants on their reasons for consuming SS, it was observed that 82.33% consumed it to improve performance, 31.47% to take care of their health, and 21.98% for both reasons. The rest of the answers referred to alleviating dietary deficits, health issues, or necessity. As for the moment in which the athletes usually consumed SS, we found that most athletes (64.22%) consumed them during training and competitions.

With respect to the main individuals who motivated the triathletes to consume SS, the main motivator was the dietitian-nutritionist (D-N) (41.6%), followed by the coach (40.7%), teammates (23.8%), and the physical trainer (16%). However, within each competitive level, these data varied. Among the regional level triathletes, the main motivator was the coach (40.8%), followed by the D-N (38.5%); at the national level, the main one was the D-N (46.7%), followed by the coach (42.7%); at the international level, the main one was the D-N (42.3%), followed by the coach and teammate (both 34.6%). With regard to the place of purchase of the SS, it was observed that triathletes purchased them mostly online (60.6%), followed by specialized stores (45.9%), with these values being very similar within all the competitive levels analyzed. Table 2 shows the mean and standard deviation of the total sample, with the sample segmented based on sex and on the level of competition.

Table 3 shows the results of the ANOVA of the SS consumed according to the different categories established by the AIS [14] and based on sex, level of competition, and their interaction in addition to the mean and standard deviation of the latter. Regarding sex, no significant differences were found in any consumption of supplements (F = 0.203–2.651; *p* = 0.105–0.652). According to the level of competition, significant differences were observed for the total sports supplements (F = 3.919; *p* = 0.021), for the total supplements from Group A (F = 4.477; *p* = 0.012), and for the ergogenic aids (F = 5.801; *p* = 0.003) belonging to this last group of supplements. Finally, in the interaction between sex and competition level, the only significant difference found was in the consumption of ergogenic aids (F = 3.067; *p* = 0.049) belonging to the Group A supplements.

Table 4 shows the pair-wise comparison between the different categories after the Bonferroni adjustment of the measured variables with significant differences depending on sex, level of competition, and their interaction. Regarding the level of competition, significant differences were observed when the regional/autonomous level was compared with the national level (*p* = 0.029–0.002) in both the total number of sports supplements and the ergogenic aids and the total number of Group A supplements consumed, with the consumption at the national level being higher in all cases. Finally, in the interaction between sex and the level of competition, significant differences were only found when comparing women at the national level with those at the regional level (*p* = 0.002), with the latter having a lower consumption of ergogenic aids.

Table 5 shows those supplements that were consumed by more than 10% of the sample, based on sex and level of competition. It was observed that bars, drinks, sports gels, and caffeine are the most-consumed supplements by the sample (83.6%, 74.1%, 61.2%, and 46.6%, respectively). Regarding the significant differences based on sex, it was found that women consumed more iron than men (*p* = 0.001) but less caffeine than men (*p* = 0.0017). Regarding the level of competition, it was observed that international level triathletes consumed more sports gels (*p* = 0.047), while national level triathletes consumed more whey protein (*p* = 0.005) and glutamine (*p* = 0.019).

## 4. Discussion

The main objective of this study was to describe the consumption pattern of different SS considering the possible differences according to sex and competitive level of a sample of triathletes. From the total sample, 92.2% declared consuming SS, with this value being very similar to the results obtained in endurance athletes such as mountain runners (93.84%) [20], although it was higher than open-water swimmers (79.5%) [22] and in other sports such as fencing, sailing, or tennis, where a lower prevalence of supplement consumption was described (46.9, 52.4, and 61.4, respectively) [17,29,30]. These differences may be due to the different physiological requirements between sports and to the different feeding opportunities in training and competitions, as in the case of open-water swimming, in which opportunities are determined by the distribution of the test [22]. Both trail running and triathlon are highly demanding sports in terms of energy substrates and water and sodium requirements. Hence, both sports show a similar consumption of SS, especially when its objective is to cover energy requirements and replace the loss of water and electrolytes [8,12,13]. In addition, a difference in SS consumption was found between triathletes at different competitive levels, with the number of SS consumed being higher in athletes from a medium-high competitive level when compared with the basic level, with these results similar to those previously found in studies with different Spanish athletes. Although some studies have analyzed the prevalence of SS consumption in endurance sports, the present one is one of the first studies on triathletes with a representative sample of 232 individuals.

In this population of triathletes, the main reason for consuming SS was to improve sports performance, followed by health care, which coincides with what was observed in other studies [16,17,19,29,30,31]. Regarding the person who advised the triathlete to consume SS, the D-N health professional was the most consulted by athletes ahead of coaches and colleagues to guide their intake of SS. This result is contrary to that obtained in other studies [16,17,27,29,32], which indicated that the person who determined SS consumption was not an expert in this field. It is important to consider that many athletes decide to use SS as part of their nutritional strategy and obtain advice, on most occasions, from personnel who are unqualified in sports nutrition. In this sense, athletes who receive advice from a D-N tend to have better eating habits, which include periodization of nutrition and consumption of SS, with broad scientific support in terms of performance improvement [33,34].

Regarding the days in which SS were usually consumed, the triathletes reported consuming them mostly both during training and in competitions (64.2%), with these values being higher than that found in previous studies in mountain runners (44.4%) [20] and open-water swimmers (39.4%) [22], which could be explained by the frequency and type of training and the high physiological demand (carbohydrates, water, and electrolytes) imposed by this discipline [5,8]. In this sense, the nutritional needs of triathletes during training or racing could be covered with the intake of water, sodium, and carbohydrates in the shape of SS. In fact, the current guidelines in sports nutrition for these types of sports disciplines recommend an intra-competition/training intake of 500 mL of liquids, 300–600 mg of sodium, and 60 to 120 g of carbohydrates per hour [12,35,36]. In addition, the post-exercise nutritional recovery is performed through the intake of carbohydrates and proteins (0.8 g carbohydrate/kg body weight plus 0.2–0.4 g protein/kg body weight) [12,35], with recovery shakes and protein powder being the most-consumed supplements after bars, sports drinks, gels, and caffeine. To put these recommendations into practice, the type of competition must be taken into account (distance, orography, expected weather, supplies, etc.) as well as the interpersonal variables of the triathletes (age, years of experience, food tolerance, sports equipment, etc.) [37,38].

On the other hand, the place where the athletes buy the SS could determine their correct usage [39,40,41]. Athletes declared a greater purchase of SS through the Internet, followed by specialized stores and pharmacies. It should be noted that purchasing SS online can be problematic and risky, as the possibility of obtaining lower-quality or even illegal SS increases due to the possible lack of specific legislation in the country of origin [40,41,42].

Statistically significant differences were found in the consumption of supplements from Group A of the AIS classification [14], according to the competitive level, for the total SS (F = 3.919, *p* > 0.021), for the total group A (F = 4.477, *p* < 0.012), and specifically for ergogenic aids (F = 5.801, *p* < 0.03), with the national-competition-level group having the highest consumption of ergogenic aids (1.17 ± 0.13 national; 0.88 ± 0.18 international; 0.80 ± 0.09 regional/autonomous). However, these differences were only significant when comparing the regional level with the national level. Similarly, statistically significant differences were found in the interaction with sex and competitive level (F = 3.067, *p* < 0.049) for the category of ergogenic aids, with national-level women consuming the most supplements in this subgroup (1.35 ± 0.21) compared to men in the same category (1.10 ± 0.14), with these differences proving significant when comparing women at regional and national competitive levels. These results coincide in part with those observed in other studies, where Group A was the most-consumed SS group and the general trend being that the higher competitive level, the greater the consumption of sports food, medical supplements, and ergogenic aids [16,21,27,31]. The fact that this sample of triathletes did not follow this trend could be explained by the % of individuals who consulted the D-N to plan the intake of supplements in each competitive category, with national-level athletes consulting the D-N the most, followed by the international athletes, and lastly, regional athletes. In addition, the international athletes declared a greater purchase of SS through the Internet, followed by regional and national athletes. These results support the importance of having qualified people such as the D-N to provide advice on the consumption of SS with a broad support from scientific evidence and the risks of buying SS over the Internet (previously discussed), which could be influencing the consumption of SS according to the competitive level [40,43].

In terms of average supplement consumption within each AIS group, Group A had the highest average consumption of supplements, with 6.05 ± 0.24 compared to Groups B (0.53 ± 0.06) and C (1.68 ± 0.15). However, it should be noted that the high consumption within group A was mainly due to the high consumption of supplements in the sports food subgroup (4.4 ± 0.17), with less consumption reported for the other subgroups within this group.

It should be noted that the four most-consumed supplements were found within Group A of the AIS classification, indicating a high degree of current scientific evidence [14]. In fact, sports bars, sports drinks, sports gels, and caffeine were the most-consumed supplements by triathletes, as these SS are very important when it comes to providing energy in the form of carbohydrates, preventing dehydration and hyponatremia during training or competitions, and for recovery after these sessions [13,44,45]. These results are similar to those obtained in previous research, which showed that in general, the most-consumed SS were isotonic or sports drinks, energy bars, and gels [16,17,19,24,29,30,31].

If we analyze the results according to sex, we find that on one hand, the four most-consumed SS coincided in both sexes, and the consumption of this type of SS, such as sports drinks, sports gels, and sports bars, was before or during training and/or competition, while the protein powders, protein bars, and mixed macronutrient supplement groups were used after training and/or competition, with their consumption contributing towards reaching the recommended nutrient intakes. On the other hand, in medical supplements within Group A, statistically significant differences were found in iron consumption when comparing it according to sex, with this consumption being higher in women, which would be justified by the high prevalence of anemia in female athletes [46] and the use of iron SS to prevent anemia in female athletes. [46]. In the group of ergogenic aids, caffeine was the most-consumed SS, with this consumption being higher in men than in women. Given that it is one of the SS with the most scientific evidence, its use is usually common in endurance sports [20,22]. According to the EFSA, the consumption of 3 mg/kg of caffeine one hour before exercising improves performance in endurance sports, and if the dose is increased to 4–6 mg/kg, the feeling of tiredness is reduced [47].

Sports performance is influenced by various factors, and athletes usually present an increase in body temperature and sweat rate when practicing sports, which leads to an increased loss of fluids and electrolytes [13,45]. These factors justify that the sample presented a high consumption of sports bars and drinks, as water imbalances during training or competitions can lead to a decrease in performance [35,45,48]. Sports foods (sports gels, sports drinks, and sports bars) provide athletes with energy and nutrients in a simpler way than conventional foods, which is why they serve as support in situations of high sports demand or at times when there is difficulty in accessing conventional food [9]. Sports drinks are a good option for replenishing fluids, electrolytes, and carbohydrates during and after exercise [49], and sports bars can also be useful during and after exercise, as they provide carbohydrates, proteins, and micronutrients [9,48,49,50].

A key piece of information found in the present study was that the consumption of SS from Group C (supplements that have little or no evidence of beneficial effects) was 1.73 ± 0.25, double the amount of consumption of SS from Group B (supplements that have some benefit, but more research is needed), which was 0.63 ± 0.5, with similar results also observed in previous studies with mountain runners [21]. This should be considered when dealing with athletes in nutrition consultations, and it is also an interesting research topic in nutrition education programs [33,40,41].

### Limitations

The questionnaire used to evaluate SS consumption was a validated tool, although one of its limitations was that the consumption information was collected in a self-reported and retrospective manner based on the memory of the triathletes. This could lead to errors in the number and type of declared SS. In general, athletes tend to worry about their diet and training, as their performance depends on it, and they are usually knowledgeable about their eating habits. This could make them better recall their food intake and SS as compared to the general population. This is one of the first studies to use a significant sample in this type of population. As future research lines, efforts can be made to collaborate with federations in other countries, to obtain a representative SS consumption pattern worldwide, and to verify whether SS consumption is similar in all of them according to the competitive level and sex.

## 5. Conclusions

The consumption of supplements by triathletes was high when compared to other sports disciplines, perhaps due to the high physiological requirements during training and competition. At a competitive level, the consumption of SS was different at the national-international level when compared to the regional level. In addition, differences were found in the consumption of some supplements according to sex. The most-consumed SS by triathletes were bars, sports drinks, sports gels, and caffeine, all included in Group A, the category with the most scientific evidence to date. Similarly, a large part of the athletes obtained advice from a D-N, which would explain why most of the SS consumed by athletes were part of Group A as well as the differences in SS consumption at different competitive levels.

## Figures and Tables

**Table 1 nutrients-15-01330-t001:** Descriptive data of the sports supplements consumed in the different categories established by the AIS [14], based on sex and level of competition.

Competitive Level	Sex	Age (Year)	Height (cm) *	Body Mass (kg) *	BMI *	Experience (Years)
Regional	Male (*n* = 97)	36.38 ± 10.18	177.56 ± 6.36	72.83 ± 9.26	23.06 ± 2.26	5.18 ± 3.33
	Female (*n* = 34)	36.35 ± 7.99	164.68 ± 6.29	59.68 ± 9.29	21.97 ± 2.74	3.82 ± 2.35
National	Male (*n* = 52)	33.37 ± 9.52	176.73 ± 5.80	68.73 ± 7.23	21.97 ± 1.73	5.12 ± 3.18
	Female (*n* = 23)	28.81 ± 10.72	163.96 ± 5.51	55.46 ± 5.94	20.62 ± 1.81	5.09 ± 3.07
International	Male (*n* = 16)	34.69 ± 9.51	176.75 ± 6.77	74.94 ± 10.48	24.04 ± 3.68	3.31 ± 2.80
	Female (*n* = 10)	34.00 ± 10.89	163.60 ± 6.83	59.60 ± 5.46	22.24 ± 1.11	4.30 ± 3.71

Data presented as M ± SD; BMI, body mass index; * height and weight are self-reported; the BMI was calculated based on that data.

**Table 2 nutrients-15-01330-t002:** Descriptive data of the sports supplements consumed in the different categories established by the AIS [14], based on sex and level of competition.

		Sex		Competitive Level			Total
		Male	Female	Regional	National	International	
Total SS		8.39 ± 0.75	8.28 ± 0.47	7.37 ± 0.44	9.39 ± 0.83	9.92 ± 1.33	8.31 ± 0.40
Group A	Sport foods	4.28 ± 0.33	4.44 ± 0.19	4.03 ± 0.19	4.83 ± 0.33	5.00 ± 0.57	4.40 ± 0.17
	Medical supplements	0.94 ± 0.14	0.60 ± 0.07	0.59 ± 0.08	0.88 ± 0.12	0.73 ± 0.20	0.70 ± 0.07
	Ergogenic aids	0.76 ± 0.13	1.00 ± 0.08	0.80 ± 0.09	1.17 ± 0.13	0.88 ± 0.18	0.93 ± 0.07
	Total	6.03 ± 0.50	6.05 ± 0.28	5.45 ± 0.27	6.88 ± 0.50	6.65 ± 0.75	6.05 ± 0.24
Group B		0.63 ± 0.12	0.50 ± 0.07	0.48 ± 0.07	0.48 ± 0.11	0.96 ± 0.25	0.53 ± 0.06
Group C		1.73 ± 0.25	1.66 ± 0.19	1.38 ± 0.17	2.00 ± 0.31	2.27 ± 0.89	1.68 ± 0.15
Group D		0.00 ± 0.00	0.00 ± 0.00	0.00 ± 0.00	0.00 ± 0.00	0.00 ± 0.00	0.00 ± 0.00

Data presented as M ± SD; SS, sport supplements.

**Table 3 nutrients-15-01330-t003:** ANOVA of the sports supplements consumed in the different categories established by the AIS [14], based on sex, level of competition, and their interaction.

		Sex		Competitive Level		Sex × Competitive Level							
Variable		F	*p*	F	*p*	Regional		National		International		F	*p*
						Male	Female	Male	Female	Male	Female		
Total SS		0.394	0.531	3.919	0.021	7.51 ± 0.61	6.97 ± 1.03	8.81 ± 0.83	10.70 ± 1.25	11.19 ± 1.50	7.90 ± 1.90	1.804	0.167
Group A	Sport foods	0.708	0.401	2.589	0.077	4.07 ± 0.26	3.91 ± 0.43	4.86 ± 0.35	4.74 ± 0.53	5.31 ± 0.63	4.50 ± 0.80	0.185	0.831
	Medical supplements	2.651	0.105	2.776	0.059	0.54 ± 0.10	0.73 ± 0.17	0.67 ± 0.14	1.35 ± 0.20	0.75 ± 0.25	0.70 ± 0.31	1.662	0.192
	Ergogenic aids	2.025	0.156	5.801	0.003	0.94 ± 0.10	0.41 ± 0.17	1.10 ± 0.14	1.35 ± 0.21	1.06 ± 0.25	0.60 ± 0.32	3.067	0.049
	Total	0.203	0.652	4.477	0.012	5.58 ± 0.37	5.09 ± 0.63	6.61 ± 0.51	7.48 ± 0.76	7.12 ± 0.92	5.90 ± 1.16	0.980	0.377
Group B		0.309	0.579	2.689	0.070	0.48 ± 0.09	0.47 ± 0.16	0.36 ± 0.13	0.74 ± 0.19	1.00 ± 0.23	0.90 ± 0.29	1.073	0.344
Group C		0.972	0.325	2.301	0.102	1.37 ± 0.23	1.41 ± 0.39	1.79 ± 0.32	2.48 ± 0.48	3.00 ± 0.58	1.10 ± 0.73	2.810	0.062
Group D		-	-	-	-	-	-	-	-	-	-	-	-

Data presented as M ± SD; SS, sport supplements.

**Table 4 nutrients-15-01330-t004:** Post hoc comparison between the variables with significant differences.

	Variable		Comparison between Groups		M ± SD	*p*	CI 95%
Competitive level	Total SS		Regional–National		−2.51 ± 0.96	0.029	−4.830 to −0.187
			Regional–International		−2.30 ± 1.35	0.271	−5.562 to 0.961
			National–International		0.21 ± 1.43	1.000	−3.234 to 3.650
	Group A	Ergogenic aids	Regional–National		−0.55 ± 0.16	0.002	−0.936 to −0.159
			Regional–International		−0.16 ± 0.23	1.000	−0.70 to 0.39
			National–International		0.39 ± 0.24	0.310	−0.185 to 0.967
		Total	Regional–National		−1.71 ± 0.59	0.012	−3.129 to −0.299
			Regional–International		−1.18 ± 0.82	0.462	−3.128 to 0.809
			National–International		0.53 ± 0.87	1.000	−1.564 to 2.633
Sex × Competitive level	Group A	Ergogenic aids	Male	Regional–National	−0.16 ± 0.18	1.000	−0.585 to 0.269
				Regional–International	−0.12 ± 0.28	1.000	−0.795 to 0.546
				National–International	0.03 ± 0.29	1.000	−0.677 to 0.744
			Female	Regional–National	−0.94 ± 0.26	0.002	−1.567 to −0.305
				Regional–International	−0.19 ± 0.34	1.000	−1.029 to 0.653
				National–International	0.75 ± 0.36	0.126	−0.138 to 1.633

SS, sport supplements.

**Table 5 nutrients-15-01330-t005:** Distribution (%) of the most-consumed supplements based on sex and competitive level according to the categories established by the AIS [14].

Category		Supplement	Total (%)	Sex (%)			Competitive Level (%)			
				M	F	*p*	R	N	I	*p*
Group A	Sport foods	Recovery shake	30.6	30.3	31.3	0.876	26.0	36.0	38.5	0.210
		Sports bars	83.6	86.1	77.6	0.115	83.2	85.3	80.8	0.847
		Protein bars	40.9	40.6	41.8	0.868	43.5	37.3	38.5	0.661
		Sports drinks	74.1	77.0	67.2	0.122	73.3	74.7	76.9	0.920
		CH gainers	19.0	20.6	14.9	0.317	16.0	21.3	26.9	0.354
		Sports confectionery	22.0	22.4	20.9	0.799	21.4	25.3	15.4	0.555
		Electrolytes	27.2	27.3	26.9	0.950	23.7	30.7	34.6	0.367
		Sports gels	61.2	59.4	65.7	0.374	55.7	64.0	80.8	0.047
		Maltodextrin	13.8	14.5	11.9	0.602	10.7	20.0	11.5	0.165
		Whey protein	29.7	27.9	34.3	0.330	21.4	42.7	34.6	0.005
		Vegetal protein	14.2	12.7	17.9	0.306	13.0	13.3	23.1	0.389
	Medical supplements	Iron	17.7	12.1	31.3	0.001	13.0	22.7	26.9	0.091
		Multivitamins	20.7	21.8	17.9	0.505	18.3	28.0	11.5	0.121
		Vitamin D	18.1	15.2	25.4	0.067	15.3	24.0	15.4	0.273
	Ergogenic aids	B alanine	13.8	14.5	11.9	0.602	11.5	17.3	15.4	0.484
		Caffeine	46.6	51.5	34.3	0.017	42.0	53.3	50.0	0.271
		Creatine monohydrate	27.2	30.9	17.9	0.044	22.9	36.0	23.1	0.112
Group B		Antioxidants/vitamin C	15.1	12.7	20.9	0.115	12.2	14.7	30.8	0.054
		Carnitine	11.6	12.1	10.4	0.719	10.7	12.0	15.4	0.787
		Collagen	12.5	11.5	14.9	0.477	15.3	4.0	23.1	0.014
Group C		BCAA	23.7	23.6	23.9	0.968	19.1	32.0	23.1	0.110
		Glutamine	19.4	20.0	17.9	0.715	13.0	28.0	26.9	0.019
		Magnesium	24.6	23.6	26.9	0.605	22.9	22.7	38.5	0.218

M, male; F, female; R, regional; N, national, I, international; BCAA, branched-chain amino acids.

## Data Availability

The data presented in this study are available in the tables of this article. The data presented in this study are available on request from the corresponding author.

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
