# Peer review of "Do Spanish Triathletes Consume Sports Supplements According to Scientific Evidence? An Analysis of the Consumption Pattern According to Sex and Level of Competition"

_nutrients, 2023, doi:10.3390/nu15061330_

Round 1
Reviewer 1 Report
This passage is discussing a study that analyzed the use of sports supplements (SS) among Spanish triathletes. The researchers found that 92.2% of the athletes consumed SS, with no significant differences between sexes or competition levels. However, they did find differences in the types of supplements consumed based on competition level, with more elite athletes consuming more supplements with stronger scientific evidence. The most commonly consumed SS were bars, sports drinks, sports gels, and caffeine. Overall, the study suggests that SS consumption is common among triathletes, particularly at higher competition levels. This manuscript is well-written, very clearly presented and the study format impeccable .
I have only three very minor comments.
1. In the 5th paragraph of the introduction, the sentence with "... nutritional intake in this ..." should be changed to these as more than one sport is being discussed.
2. I am curious why standard error rather than standard error of the mean is being used. Since all participants are biological replicates, it would seem reasonable to use SEM and perhaps you might see some statistically significant differences that you could report compared to using SD. I defer to the authors, but wanted to comment.
3. In the results, the consumption was similar between all three groups with no significant differences. Might the author consider tailoring the sentence with "... although the percentages of consumption in all three categories were similar, there was an increased trending between regional, national and international athletes.
Author Response
- In the 5th paragraph of the introduction, the sentence with "... nutritional intake in this ..." should be changed to these as more than one sport is being discussed.
Response of the authors: We appreciate the reviewer's comments and thank the interest in the work done, the error in the phrase has been corrected in the manuscript.
- I am curious why standard error rather than standard error of the mean is being used. Since all participants are biological replicates, it would seem reasonable to use SEM and perhaps you might see some statistically significant differences that you could report compared to using SD. I defer to the authors, but wanted to comment.
Response of the authors: We appreciate the reviewer's comment, we have used the standard error instead of the standard error of the mean because in previous investigations of the same subject the standard error has been used and we wanted to follow the same statistical treatment of the data (DOI), as you can see in the articles included below. In case you find it more interesting to use the standard error of the mean, we would appreciate the suggestion and we would make the change.
- https://doi.org/10.3390/nu14245211
- https://doi.org/10.3390/nu15020262
- https://doi.org/10.3390/nu12113357
- https://doi.org/10.3390/nu10101341
- https://doi.org/10.1186/s13102-021-00278-0
- https://doi.org/10.3390/nu12040993
- In the results, the consumption was similar between all three groups with no significant differences. Might the author consider tailoring the sentence with "... although the percentages of consumption in all three categories were similar, there was an increased trending between regional, national and international athletes.
Response of the authors: We appreciate the reviewer's comment, the proposed sentence has been included for further information.
Reviewer 2 Report
I have no comments as it is very well written, only english needs to be checked.
Author Response
I have no comments as it is very well written, only english needs to be checked.
Response of the authors: We appreciate the reviewer's comment and we are looking forward the to work be published. A native English speaker has reviewed the manuscript
Reviewer 3 Report
This manuscript investigated the pattern of sports supplements (SS) consumption by triathletes according to sex and the competitive level by descriptive-cross-sectional study of 232 Spanish federated triathletes. The results found no significant differences were found in terms of competition level or sex, while the manuscript still has some important points, and the results are readable. It may be of potential interest to the reader. However, there are some flaws in the manuscript that need to be revised before publishing.
1. The title “Do triathletes use sports supplements according to scientific evidence? An analysis of the consumption pattern according to sex and level of competition.” Should revise to be more concise and focus on the topic.
2. For the introduction part, maybe revise to be more concise, focus on the research review on factors that affects the sports supplements, as well as how sports supplements affect sports performance? And why the author chose sex or competition level to analyse?
Author Response
- The title “Do triathletes use sports supplements according to scientific evidence? An analysis of the consumption pattern according to sex and level of competition.” Should revise to be more concise and focus on the topic.
Response of the authors: We appreciate the reviewer's comment, the authors have changed the title.
- For the introduction part, maybe revise to be more concise, focus on the research review on factors that affects the sports supplements, as well as how sports supplements affect sports performance? And why the author chose sex or competition level to analyse?
Response of the authors: Following the reviewer's suggestions, the authors have made modifications in the introduction section to include the main mechanisms/benefits of the most consumed supplements in triathlete’s performance. Also, a sentence has been added with the justification for the choice of sex and competitive level.